# Can Circulating Cardiac Biomarkers Be Helpful in the Assessment of *LMNA* Mutation Carriers?

**DOI:** 10.3390/jcm9051443

**Published:** 2020-05-12

**Authors:** Przemyslaw Chmielewski, Ewa Michalak, Ilona Kowalik, Maria Franaszczyk, Malgorzata Sobieszczanska-Malek, Grazyna Truszkowska, Malgorzata Stepien-Wojno, Elzbieta Katarzyna Biernacka, Bogna Foss-Nieradko, Michal Lewandowski, Artur Oreziak, Maria Bilinska, Mariusz Kusmierczyk, Frédérique Tesson, Jacek Grzybowski, Tomasz Zielinski, Rafal Ploski, Zofia T. Bilinska

**Affiliations:** 1Unit for Screening Studies in Inherited Cardiovascular Diseases, National Institute of Cardiology, 04-628 Warsaw, Poland; pchmielewski@ikard.pl (P.C.); emichalak@ikard.pl (E.M.); mstepien@ikard.pl (M.S.-W.); bfn@ikard.pl (B.F.-N.); 2Department of Coronary Artery Disease and Cardiac Rehabilitation, National Institute of Cardiology, 04-628 Warsaw, Poland; ikowalik@ikard.pl; 3Molecular Biology Laboratory, Department of Medical Biology, National Institute of Cardiology, 04-628 Warsaw, Poland; m.fran@wp.pl (M.F.); gtruszkowska@ikard.pl (G.T.); 4Department of Heart Failure and Transplantology, National Institute of Cardiology, 04-628 Warsaw, Poland; m.sobieszczanska@ikard.pl (M.S.-M.); tzielin@ikard.pl (T.Z.); 5Department of Congenital Heart Diseases, National Institute of Cardiology, 04-628 Warsaw, Poland; e.biernacka@ikard.pl; 62nd Department of Arrhythmia, National Institute of Cardiology, 04-628 Warsaw, Poland; mlewan@ikard.pl; 71st Department of Arrhythmia, National Institute of Cardiology, 04-628 Warsaw, Poland; aoreziak@ikard.pl (A.O.); mbilinska@ikard.pl (M.B.); 8Department of Cardiac Surgery and Transplantology, National Institute of Cardiology, 04-628 Warsaw, Poland; m.kusmierczyk@ikard.pl; 9Interdisciplinary School of Health Sciences, Faculty of Health Sciences, University of Ottawa, Ottawa, ON K1N 6N5, Canada; ftesson@uOttawa.ca; 10Department of Cardiomyopathy, National Institute of Cardiology, 04-628 Warsaw, Poland; j.grzybowski@ikard.pl; 11Department of Medical Genetics, Medical University of Warsaw, 02-106 Warsaw, Poland; rploski@wp.pl

**Keywords:** laminopathy, *LMNA*, biomarkers, troponin T, NT-proBNP, malignant ventricular arrhythmia, arrhythmic risk stratification

## Abstract

Mutations in the lamin A/C gene are variably phenotypically expressed; however, it is unclear whether circulating cardiac biomarkers are helpful in the detection and risk assessment of cardiolaminopathies. We sought to assess (1) clinical characteristics including serum biomarkers: high sensitivity troponin T (hsTnT) and N-terminal prohormone brain natriuretic peptide (NT-proBNP) in clinically stable cardiolaminopathy patients, and (2) outcome among pathogenic/likely pathogenic lamin A/C gene (*LMNA)* mutation carriers. Our single-centre cohort included 53 patients from 21 families. Clinical, laboratory, follow-up data were analysed. Median follow-up was 1522 days. The earliest abnormality, emerging in the second and third decades of life, was elevated hsTnT (in 12% and in 27% of patients, respectively), followed by the presence of atrioventricular block, heart failure, and malignant ventricular arrhythmia (MVA). In patients with missense vs. other mutations, we found no difference in MVA occurrence and, surprisingly, worse transplant-free survival. Increased levels of both hsTnT and NT-proBNP were strongly associated with MVA occurrence (HR > 13, *p* ≤ 0.02 in both) in univariable analysis. In multivariable analysis, NT-proBNP level > 150 pg/mL was the only independent indicator of MVA. We conclude that assessment of circulating cardiac biomarkers may help in the detection and risk assessment of cardiolaminopathies.

## 1. Introduction

Dilated cardiomyopathy (DCM) is a major cause of heart failure (HF) and has a genetic basis in 40–50% of cases [1]. There is growing evidence of distinct arrhythmogenic phenotype in DCM related to *LMNA*, *SCN5A*, *PLN*, *RBM20*, *FLNC*, and *DSP* mutations [2,3,4,5,6]. Of these, cardiolaminopathies have been studied most extensively and are phenotypically quite well characterized [7,8,9,10,11,12,13]. Nevertheless, none of the studies involved baseline characteristics including circulating cardiac biomarkers. In 2005, a Canadian–Irish–Polish joint study showed the presence of *LMNA* mutations in 4.4% of consecutive DCM cases [14]. In a further study, we found that 7.6% of 66 heart transplant (HTX) recipients and 9.1% of consecutive DCM patients referred for familial evaluation carry *LMNA* mutations [15]. Since then, we have identified subsequent 28 *LMNA* mutation carriers in the National Institute of Cardiology, Warsaw.

In this study, we sought to assess the clinical characteristics including serum biomarkers, penetrance of abnormal clinical findings, and prognostic risk factors in all identified *LMNA* mutation carriers.

## 2. Materials and Methods

All carriers signed a written informed consent form for the genotyping and consented to the publishing of all data generated. This study was funded by external grant 0010/P05B/98/14 from the Polish Committee for Scientific Research, statutory grants from the National Institute of Cardiology (Warsaw, Poland) no 2.57/VII/03, 2.18/II/08, 2.10/II/10, and 2.56/II/14; external grant from National Science Centre Poland 2011/01/B/NZ4/03455 RP; and the recent NCBiR ERA-CVD DETECTIN-HF/2/2017 IB.4/II/17 grant. The study received the approval of the Bioethics Committee of the National Institute of Cardiology.

Data from all persons (probands and relatives) with *LMNA* mutations and cardiac involvement in the care of the National Institute of Cardiology, Warsaw, were retrospectively collected. All mutations were identified between 2000 and 2018 and were considered to be pathogenic/likely pathogenic according to The American College of Medical Genetics and Genomics (ACMG) criteria [16]. On a prospective basis, genetic testing was offered to all probands and all agreed to participate in the study. Cascade screening was offered to all probands’ families. Baseline clinical information from the first documented visit to the Institute and follow-up data were recorded. In particular, data were obtained for all major cardiovascular events. The probands as well as their informed and consenting relatives underwent a clinical examination, 12-lead electrocardiography, two-dimensional Doppler echocardiography, 24-h Holter ECG monitoring, and blood sampling for genetic testing. In all probands, coronary angiography or, more recently, coronary computed tomography angiography was performed. In addition, whenever available, we quantified the serum biomarkers—high sensitivity troponin T (hsTnT) and N-terminal prohormone brain natriuretic peptide (NT-proBNP)—at baseline and during ambulatory visits in patients without worsening of HF for 3 months.

In patients with no history of sudden cardiac arrest (SCA) or sustained ventricular tachycardia (sVT), we evaluated the prognostic value of circulating cardiac biomarker concentrations, assessed during initial visit or in the time window ± 6 months with regard to occurrence of malignant ventricular arrhythmia (MVA) during the follow-up, and compared it with other established risk factors. We also sought to examine the influence of such factors as proband status, sex, and type of mutation on life-time prognosis with regard to occurrence of end-stage HF and MVA.

### 2.1. Definitions

Left ventricular enlargement (LVE) was ascertained when the left ventricular end diastolic diameter (LVEDD) exceeded 112% of the predicted value, corrected for age and body surface area according to Henry’s formula, while left ventricular dysfunction (LVD) was ascertained when left ventricular ejection fraction (LVEF) was <50%. The diagnosis of DCM was made when both criteria were met. When no LVE but more distinct LVD was present (LVEF < 45%), hypokinetic non-dilated cardiomyopathy (HNDC) was diagnosed [17]. In the presence of other relevant abnormalities, such as LVE > 117%, cardiac conduction defect (CCD), or atrial or ventricular arrhythmias unexplained by other conditions, we used the term indeterminate cardiomyopathy (indeterminate CM).

CCD included atrioventricular block (AVB) and left bundle branch block (LBBB). First-degree AVB was defined by a PR interval >200 ms on standard 12-lead ECG. High-degree AVB included type II second degree or third degree AVB. Atrial arrhythmias included atrial fibrillation, flutter, and paroxysmal tachycardia lasting ≥30 s. Non-sustained ventricular tachycardia (nsVT) was defined as ≥3 consecutive ventricular beats at >120 bpm and a duration for <30 s on 24-h Holter electrocardiographic monitoring. If the VT lasted over 30 s, it was considered sustained (sVT). Ventricular arrhythmias included sVT, nsVT, and frequent ventricular extrasystoles (>500/24 h). 

HF was recognized in the presence of typical symptoms (e.g., breathlessness or fatigue), accompanied by structural and/or functional cardiac abnormalities, resulting in a reduced cardiac output and/or elevated intracardiac pressures. End-stage HF was defined as HTX, implantation of left ventricular assist device, or death caused by HF.

MVA was defined as sudden cardiac death (SCD), cardiopulmonary resuscitation (CPR), or appropriate implantable cardioverter defibrillator (ICD) shock (an ICD discharge for termination of ventricular fibrillation/VT). Death was classified as sudden if it occurred within 1 h of the onset of cardiac manifestations, during sleep (in the absence of previous hemodynamic deterioration), or within 24 h after the patient was last seen apparently stable clinically. Sudden cardiac arrest (SCA) was defined as occurring within 1 h of the onset of acute symptoms and was reversed by CPR. Relatives included all probands’ family members with *LMNA* gene mutation confirmed as a result of cascade screening, irrespective of degree of kinship. A family history of SCD was considered positive if ≥1 first degree relative had died suddenly before the age of 60 years.

### 2.2. Biomarkers’ Measurements

The plasma levels of NT-proBNP were measured by the electrochemiluminescent immunoassays Elecsys 2010 (Roche, Mannheim, Germany). Two cutoff points of NT-proBNP were used: 125 pg/mL, upper limit of normal values in the assay defined by the manufacturer, and 150 pg/mL after analysis of a receiver-operating characteristic curve (ROC), identifying the criterion of maximum sensitivity and specificity for MVA occurrence. The plasma levels of cardiac troponin T were measured by the troponin T hs-STAT (Roche, Mannheim, Germany). Again, two cutoff points for hsTnT were used: 14 ng/L, upper limit of normal values in the assay defined by the manufacturer, and 20 ng/L, after the ROC analysis which fulfils the criterion of maximum sensitivity and specificity for MVA occurrence. All measurements were performed in the National Institute of Cardiology laboratory.

### 2.3. Mutation Screening

DNA was obtained from the peripheral blood by phenol extraction or the salting-out method. Direct Sanger sequencing of twelve *LMNA* exons (canonical transcript NM_170707.4) including flanking intronic regions was performed in 16 probands as described previously [15]. In one case, it was followed by multiplex ligation-dependent probe amplification (MRC Holland, Amsterdam, Netherlands) [18]. In 5 probands, next generation sequencing (NGS) was performed. Libraries were prepared using an Illumina: TruSeq Exome Enrichment Kit for whole exome sequencing (WES) in 1 proband, Nextera Rapid Capture Exome Library Kit with different enrichment probes for TruSight One (TSO) gene panels in 2 probands and TruSight Cardio (TSC) Sequencing Kit in 2 probands (Illumina, San Diego, CA, USA). WES and TSO libraries were sequenced on Illumina HiSeq1500 and TSC on Illumina MiSeq. Library preparation, sequencing, and data analysis were performed as described previously [19]. Variants identified in probands were followed up by Sanger sequencing of relatives’ DNA using BigDye Terminator v3.1/v1.1 Cycle Sequencing Kit (Life Technologies) according to the manufacturer’s instructions and the 3500xL/3130xL Genetic Analyzer (Life Technologies, Carlsbad, CA, USA). The results were analysed with Variant Reporter 1.1 Software (Life Technologies). Non-missense mutations included insertions, deletions, nonsense mutations, or mutations affecting splicing.

### 2.4. Statistical Analysis

All results for categorical variables were presented as numbers and percentages and, for continuous variables, as mean and standard deviation (SD) or median and quartiles (Q1:25th–Q2:75th percentiles). The Fisher exact test was used for comparison of categorical variables. The differences between continuous variables were tested by Student’s t-test (for two independent samples and for paired observation, normally distributed data) or, in the case of irregular distribution, nonparametric Mann–Whitney and paired signed rank tests. A receiver-operating characteristic curve (ROC) analysis was used to assess the cutoff point of the markers for the prediction of events (Appendix A). The optimal cutoff was defined as the value with the maximal sum of sensitivity and specificity. Event analysis over time was made by using the univariable and multiple Cox proportional-hazards regression model. In order to indicate independent predictors of events, the stepwise variable selection procedure was used. Risk was quantified as a hazard ratio with 95% confidence interval (CI). Survival curves were constructed by the Kaplan–Meier method and compared by the log-rank test. We used two different start points for the time-to-event analysis: from the first cardiologic assessment in our centre and from date of birth to assess life-time risk. In subjects without an event, the follow-up period extended to the most recent evaluation or the date of July 31st, 2019. All hypotheses were two-tailed with a 0.05 type I error. All statistical analyses were performed using SAS statistical software, version 9.4 (SAS Institute, Cary, NC, USA) and Statistica v16.

## 3. Results

### 3.1. Molecular Findings in the Study Cohort

We identified 18 different *LMNA* variants (three identified twice) in 21 probands (Table 1). Of the 18 variants, 14 were described before [13,14,15,18,20,21,22,23,24,25,26,27,28] and four were novel. The identified variants were pathogenic (*n* = 12) or likely pathogenic (*n* = 6) according to ACMG criteria. Figure 1 shows the distribution of *LMNA* variants found in this study in the topology of the *LMNA* gene. Eight (38.1%) of 21 probands carried *LMNA* missense variants. Of the 13 probands with non-missense variants, seven had nonsense variants, four had frameshift variants, one had a large deletion, and one was splice variant. Non-missense variants were expected to result in truncation of the protein or, in one case (c.640-10A>G), in aberrant splicing resulting in a three amino acid insertion [29]. In five probands which had NGS performed, no additional likely pathogenic/pathogenic rare variants according to ACMG criteria in coding/splicing regions of genes causing DCM were found.

### 3.2. Study Population and Clinical Characteristics

Figure 2 shows the diagnoses of the probands and relatives at baseline and at last follow-up. At baseline, the majority of probands had DCM (*n* = 17.81%), one (5%) HNDC, and three (14%) indeterminate CM, while among relatives, four (13%) had DCM, 18 (56%) had indeterminate CM, and ten (31%) had no signs of cardiomyopathy.

Table 2 shows baseline clinical characteristics of 21 probands and 32 relatives. In the study group, relatives were a decade younger than probands (*p* = 0.002), allowing us to better understand the early phase of the disease. As expected, HF symptoms with advancement of left ventricular dysfunction were more frequent in probands as well as arrhythmias and ICD requirements. Of interest, while AVB was present in nearly half of the relatives (43.7%), LBBB was found in probands only. In line with the previous findings, the presence of elevated NT-proBNP was significantly more common in probands than in relatives (*p* = 0.003); however, the presence of elevated levels of hsTnT merely tended to be more common in probands versus relatives (*p* = 0.065), indicating that elevated hsTnT might be a useful biomarker of the onset of the disease. Of note, elevated hsTnT level tended to be more common than elevated creatine phosphokinase (CK) activity, a widely accepted marker of syndromic form of cardiolaminopathy (48% vs. 28% of *LMNA* mutation carriers respectively, *p* = 0.06).

### 3.3. Penetrance of Cardiolaminopathy Indicators

Penetrance of cardiac disorders along with abnormal levels of cardiac biomarkers was age dependent (Figure 3). During lifespan, the earliest abnormality, emerging in the second decade of life in 12% of *LMNA* variant carriers and in the third decade in 27%, was hsTnT level >14 ng/L, followed by the presence of AVB and HF (each 5% in the 2nd and 15% in the 3rd decade of life) and MVA (2% and 13%, respectively). Penetrance of cardiolaminopathy indicators was nearly complete in the 7th decade of life with 98% of patients presenting with AVB, 100% with SVA, 90% with HF, 92% with serum biomarker hsTnT >14 ng/L, and 100% with NT-proBNP >125 pg/mL.

During the follow-up, there was a modest but significant increase in the hsTnT level (Table 3), possibly reflecting progressive myocardial damage. The change in NT-proBNP level was much more noticeable, reflecting that HF was more common and advanced at the end of the follow-up period. Changes in cardiac biomarkers’ concentrations measured during the initial and subsequent visits in patients with two or more measurements are shown in Appendix A.

### 3.4. Phenotypic Differences in Missense versus Non-Missense LMNA Variant Carriers

We assessed clinical differences among non-missense and missense variant carriers at last follow-up. Of interest, missense variant carriers had LBBB more frequently (54.5% vs. 14.3%, *p* = 0.017) and more advanced cardiomyopathy with regard to left ventricular size and function (Table 4). Of note, all subjects with elevated CK activity were found among non-missense variant carriers ( Appendix A). Other variables were comparable; in particular, we found no significant differences with regard to cardiac biomarkers’ concentrations.

### 3.5. Follow-Up and Risk Stratification Including Biomarkers

Mean follow-up was 1769 days, median 1522 (Q1: 771–Q3: 2564). During the follow-up period, 14/53 (26.4%) patients developed end-stage HF: three (5.6%) of them died, and eleven (20.8%) were transplanted. There was one SCD, while another patient had SCA with successful resuscitation.

At the end of follow-up only 19/53 (36%) patients remained free of ICD (including a patient who refused to have an ICD implanted and died suddenly). Of the 34/53 (64%) patients with ICD, 10/34 (29%) experienced adequate ICD discharge.

The disease progressed to end-stage HF only in patients with DCM/HNDC. MVA occurred in 41% of them and in 19% of patients with indeterminate status ( Appendix A).

The comparison of clinical data of probands and relatives between the initial and last follow-up visits confirmed progressive character of *LMNA*-related disease both with regard to HF and to arrhythmia (Table 2 and Appendix A).

The influence of prespecified risk factors on the risk of MVA assessed from the date of the initial evaluation in the subgroup of 44 patients with no history of SCA or sVT and, on the risk of end-stage HF and MVA during lifespan in the whole cohort of 53 patients, is summarized in the Table 5.

#### 3.5.1. Arrhythmic Risk Stratification During the Follow-Up

The univariable analysis of MVA events during follow-up showed no impact of sex and type of mutation on MVA occurrence and confirmed the involvement of established risk factors, such as the presence of AVB, nsVT, or reduced LVEF (with cutoff value at 55%).

The analysis of the influence of circulating biomarker concentrations on MVA occurrence suggested that circulating biomarker concentrations could be more potent risk factors than established risk factors: hazard ratio for hsTnT with a cutoff value at 20 ng/L was 13.2 (95% CI: 1.5–115.8, *p* = 0.020) and, for NT-proBNP with a cutoff value at 150 pg/mL, was 13.4 (95% CI: 1.6–112.7, *p* = 0.017).

In fact, in multivariable analysis, elevated NT-proBNP level was the only indicator of the occurrence of MVA at 8 years of follow-up (HR: 10.4, 95% CI: 1.21–89.79, *p* = 0.010).

The Kaplan–Meier curves showing MVA-free survival during the follow-up period according to respective risk factors are available online at Appendix A.

#### 3.5.2. Factors Affecting Lifelong Prognosis in Cardiolaminopathies

Kaplan–Meier analysis from the date of birth in the whole cohort showed a worse transplant-free survival among probands versus relatives (*p* = 0.0339) (Appendix A), male vs. female patients (*p* = 0.0041), and surprisingly in patients with missense vs. other mutations (*p* = 0.0412) (Figure 4). Analogous analysis with respect to MVA events (Figure 5 and Appendix A) showed trends toward more common MVA among males vs. females (*p* = 0.0872) and patients with missense vs. other variants (*p* = 0.0948) and no impact of the proband status (*p* = 0.7469).

In multivariable Cox regression analysis (Table 5), we found that male sex (HR: 6.18, 95% CI: 1.66–23.0, *p* = 0.007) and missense variants (HR: 3.83, 95% CI: 1.14–12.85, *p* = 0.029) were independently related with the occurrence of end-stage HF at 60 years.

## 4. Discussion

### 4.1. Penetrance of Cardiolaminopathy Indicators

A major finding of the study is that the earliest marker of the carrier status in *LMNA*-related cardiomyopathy is elevated hsTnT level, followed by widely recognized features, such as AVB, HF, and MVA [9,10,30]. In 2013, Rapezzi et al. [31] proposed a framework for the clinical approach to diagnosis in cardiomyopathies based on the recognition of diagnostic ”red flags” that can be used to guide rational selection of specialized tests. Our study shows that elevated hsTnT level > 14 ng/L was present in two thirds of probands and in more than one third of relatives, indicating that it might be a “red flag” to introduce at least lifestyle modifications in asymptomatic or mildly symptomatic carriers. To the best of our knowledge, circulating cardiac biomarkers in relation to the disease penetrance or prognosis have not been reported in cardiolaminopathies. The role of circulating cardiac biomarkers in the detection of HF has been established [32]. Little is known about the significance of circulating biomarkers in early stage of DCM in humans [33], while routinely used biomarkers (NT-proBNP and hsTnT) are widely accepted in the diagnosis of occult DCM in Doberman Pinschers [34].

In the study, similarly to others, we found high and age-dependent penetrance of cardiac manifestations in *LMNA* mutation carriers [10,21,30]. Despite the fact that our study group was young (mean age at baseline was 33.2 years for all carriers and 29.0 years for the relatives), the majority of relatives (56%) had indeterminate CM and phenotypic expression was absent in only 31% of them. In the study by Kumar et al. [10], 18/35 (51.4%) relatives were phenotypically normal while the study cohort was older than the one we studied (mean age of 41 years). In the study by Pasotti et al. [21], 29/67 (43%) relatives had no signs of cardiomyopathy. This discrepancy may be related to adopted definitions of early stage disease but may also underscore the need of diagnostic vigilance (e.g., repeated 24-h ECG examinations). Of note, the Cardiomyopathy Registry of the EURObservational Research Programme of the European Society of Cardiology [35] showed that Holter monitoring was performed only in 37% of DCM patients. In the setting of cardiolaminopathies, it is of crucial importance to monitor asymptomatic carriers with Holter monitoring; however, the appearance of elevated hsTnT concentrations precedes the appearance of arrhythmia, as shown in our study.

### 4.2. Phenotypic Differences in Missense versus Non-Missense LMNA Variants

No significant differences were found with regard to cardiac biomarkers’ levels between missense and non-missense *LMNA* variant carriers. A novel finding in this study is the higher frequency of LBBB (54.5% vs. 14.3%, *p* = 0.017) among missense versus non-missense *LMNA* carriers. It cannot however be excluded that the observed difference is dependent on the degree of left ventricular impairment rather than on the mutation type. More advanced heart failure among missense variant carriers may in turn result from the fact that probands constituted almost two thirds of this group but only one third of non-missense variant carriers (*p* = 0.063). In the study by Nishiuchi et al. [36], prevalence of low LVEF and degree of left ventricular dilation was similar in the truncation mutation vs. missense mutation group. There are scarce data on the relationship between the level of conduction defects and type of *LMNA* mutations. Nishiuchi et al. [36] showed that AVB was more common among patients with non-missense variants. In our study, the frequencies of LBBB, right bundle branch block (RBBB), and AVB at baseline were 17%, 0%, and 60%, respectively, while in the recent and largest study to date by Wahbi et al., it was lower with regard to LBBB and AVB (4.6% and 34.7%, respectively), whereas 6% of patients had RBBB. This underlines the differences in the studied populations.

### 4.3. Arrhythmic Risk Stratification Including Biomarkers

Another interesting finding of our study was the strong, independent association between NT-proBNP level > 150 pg/mL and the occurrence of MVA among *LMNA* mutation carriers.

Recently, several studies defining prognostic markers of SCD in arrhythmogenic DCM and cardiolaminopathies have been published [2,7,36,37], with European Society of Cardiology (ESC) guidelines [38] adopting the results of an earlier study by van Rijsingen et al. [7]. Based on retrospective eight-centre analysis, the authors proved that non-missense variants were independently associated with MVA. Similarly, based on retrospective 5-centre analysis, Kumar et al. [10] showed the independent association of non-missense variants with sustained ventricular arrhythmia and death. Wahbi et al. [37] presented an MVA risk prediction model for *LMNA* variant carriers and further stressed the role of non-missense variants as one of significant risk factors along with male sex, nsVT, AVB of 1st and higher degree, and reduced LVEF. In contrast, Pasotti et al. [21] did not show any association between non-missense *LMNA* variants and a worse prognosis, similarly to Captur et al. [39], who recently performed hierarchical cluster analysis of the published literature. In our cohort, we also could not confirm worse MVA-free survival in relation to *LMNA* non-missense variants either during the follow-up period or from the date of birth, and surprisingly, they were characterized by more favourable prognosis with regard to end-stage HF during lifespan. Further studies are warranted to explain the differences and to better assess the impact of the type of *LMNA* mutations.

The excellent prognostic role of NT-proBNP in patients with HF is widely recognized [40]. The association of raised levels of NT-proBNP and MVA in HF patients was shown previously [41,42]. Nevertheless, it has neither been included in currently used risk models nor has it been evaluated in *LMNA*-related cardiac disorder.

The utility of cardiac troponins in determining prognosis in HF and DCM was shown, primarily in patients with acute HF admitted to hospitals but also in ambulatory care [40,43]. To our knowledge, however, their usefulness has been investigated with regard to HF end-points, such as HF deaths and hospitalizations [43], and not as a predictor of arrhythmic risk. In our study, elevated hsTnT > 20 ng/L was associated with increased MVA occurrence only in univariable analysis; thus, its role as an independent risk factor needs further evaluation. Also, due to recent advances in cardiac biomarker sensing technologies [44], point-of-care (POC) applications can be hopefully used in the management of cardiomyopathies in the foreseeable future.

### 4.4. Molecular Findings in the Study Cohort

In our study, *LMNA* non-missense mutation carriers constituted nearly two thirds (62%) of the cohort. For comparison, in the study by Nishiuchi et al. [36], they constituted 75% of patients, whereas in the cohorts by Kumar et al. [10] and van Rijsingen et al. [7], they represented 42% and 45%, respectively. In the study by Wahbi et al. [37], carriers of non-missense variants were less common in the derivation sample (29%) but better represented in the smaller validation sample (46%).

### 4.5. Study Limitations

The major limitation of the study is small sample size due to its single-centre character and, hence, small number of major cardiovascular events, precluding the use of multivariate analysis models. The retrospective observational design of the study may include confounders. This study comes from a tertiary referral centre, one of two leading cardiological centres in Poland performing HTX. Therefore, patients referred to this tertiary referral centre may present with more severe disease than patients usually admitted in other hospitals.

## 5. Conclusions

*LMNA* mutation-related cardiac disorders are associated with high and age-dependent penetrance of cardiac manifestations, rapid progression to end-stage HF, and high incidence of life-threatening arrhythmic events. Elevated hsTnT level seems the earliest abnormality emerging in the course of cardiolaminopathies and may facilitate early detection of the *LMNA* carrier status. Circulating cardiac biomarkers, especially increased NT-proBNP level, may be helpful in arrhythmic risk stratification.

## Figures and Tables

**Figure 1 jcm-09-01443-f001:**
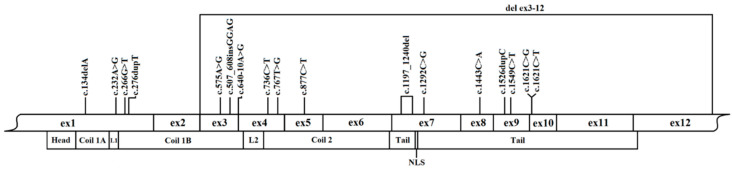
Distribution of *LMNA* variants in our study cohort. Legend: NLS, nuclear localization signal.

**Figure 2 jcm-09-01443-f002:**
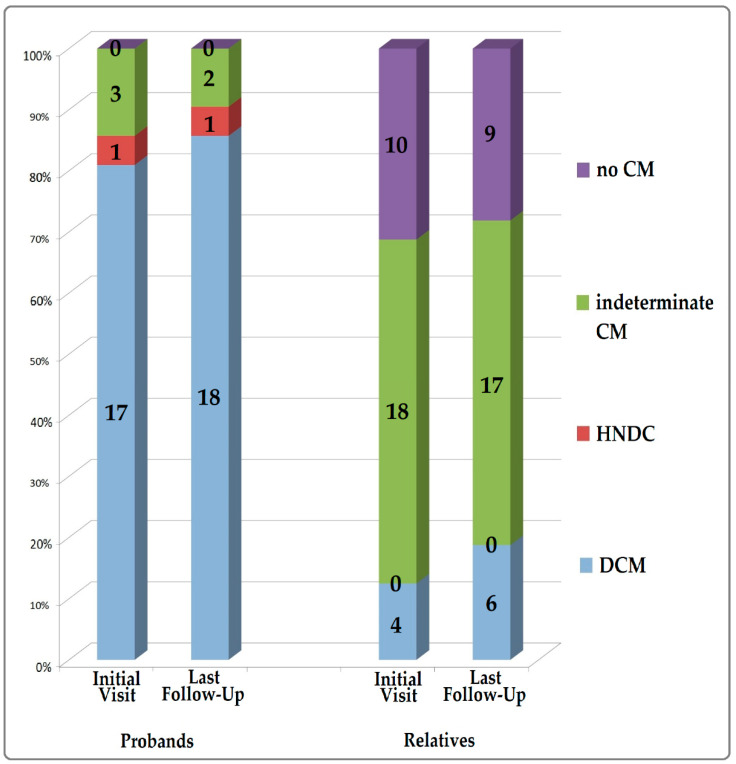
Diagnoses of probands and relatives at initial visit and last follow-up. Legend: CM: cardiomyopathy; DCM: dilated cardiomyopathy; HNDC: hypokinetic non-dilated cardiomyopathy.

**Figure 3 jcm-09-01443-f003:**
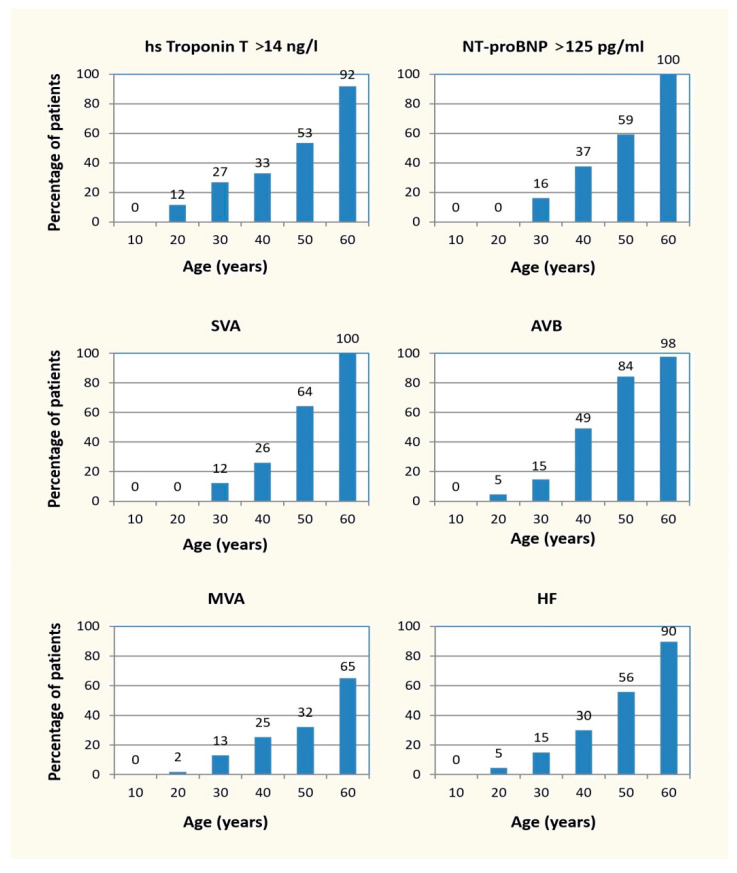
Penetrance of cardiolaminopathy indicators. Legend: AVB: atrioventricular block; HF: heart failure; hs: highly sensitive; MVA: malignant ventricular arrhythmia; NT-proBNP: N-terminal pro-brain natriuretic peptide; SVA: supraventricular arrhythmia.

**Figure 4 jcm-09-01443-f004:**
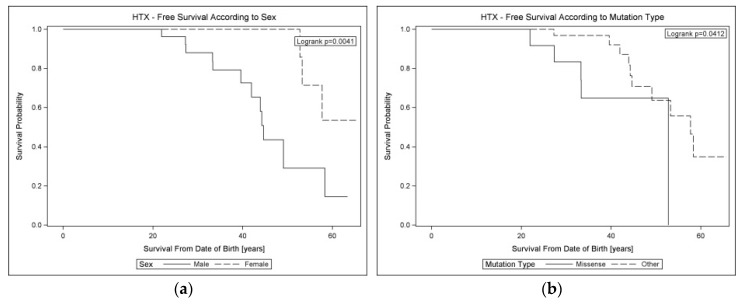
Kaplan–Meier lifelong HTX-free survival curves in cardiolaminopathy according to (**a**) sex and (**b**) mutation type. Legend: HTX: heart transplantation.

**Figure 5 jcm-09-01443-f005:**
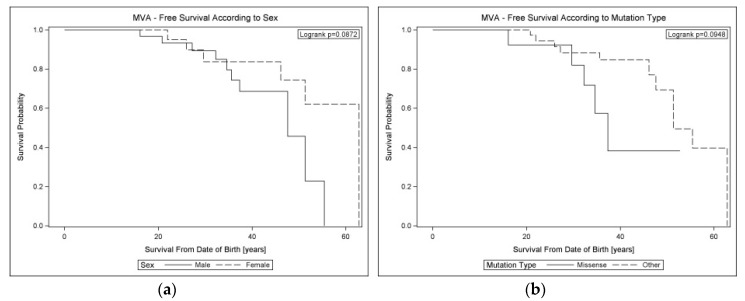
Kaplan–Meier lifelong MVA-free survival curves in cardiolaminopathy according to (**a**) sex and (**b**) mutation type. Legend: MVA: malignant ventricular arrhythmia.

**Table 1 jcm-09-01443-t001:** Genotyping results and clinical phenotypes in probands.

*LMNA* Gene Mutation	Protein Mutation	Type	Exon (NLS Class)	ACMG Classification	No. of Probands/Relatives	Proband Phenotype	References
c.134delA	p.Tyr45Ser fsTer51	truncation	1 (1)	pathogenic	1/1	HNDC, AVB, AF, nsVT, HTX	novel
c.232A>G	p.Lys78Glu	missense	1 (1)	likely pathogenic	1/1	DCM, SCA, AVB, HTX	Kourgiannidis et al. [20]
c.266G>T	p.Arg89Leu	missense	1 (1)	likely pathogenic	1/0	DCM, AVB, AF, HTX	Pasotti et al. [21], Taylor et al. [13], Saj et al. [15]
c.276dupT	p.Asp93Ter	truncation	1 (1)	pathogenic	1/3	DCM, AF, nsVT	novel
del_ex3-12	n/a	truncation	2/3 (1)	pathogenic	1/0	DCM, AVB, AF, nsVT, ICD shocks, HTX	Gupta et al. [18], Saj et al. [15]
c.575A>G	p.Asp192Gly	missense	3 (1)	likely pathogenic	2/1	DCM, AVB, AF, nsVT, HTX/HF death	Sylvius et al. [14], Saj et al. [15], Fidziańska et al. [28]
c.607_608 insGGAG	p.Glu203GlyfsTer12	truncation	3 (1)	pathogenic	1/4	DCM, AVB, sVT, HTX	novel
c.640-10A>G	n/a	inframe insertion	3/4 (1)	likely pathogenic	1/2	DCM, sVT	Otomo et al. [29], Ito et al. [27]
c.736C>T	p.Gln246Ter	truncation	4 (1)	pathogenic	1/0	DCM, AVB, AF, nsVT, HF death	Pasotti et al. [21], Saj et al. [15]
c.767T>G	p.Val256Gly	missense	4 (1)	likely pathogenic	1/0	DCM, AF, AVB, nsVT, HF death	Saj et al. [15]
c.877C>T	p.Gln293Ter	truncation	5 (1)	pathogenic	1/0	LVE, AVB, AF, nsVT	novel
c.1197_1240del	p.Gly400Arg fsTer11	truncation	7 (1)	pathogenic	1/2	DCM, SCA, ICD shocks, AVB, AF	Saj et al. [15]
c.1292C>G	p.Ser431Ter	truncation	7 (2)	pathogenic	1/5	DCM, AVB, nsVT, HTX	Saj et al. [15]
c.1443C>A	p.Tyr481Ter	truncation	8 (2)	pathogenic	1/3	DCM, AVB, AF, nsVT, HTX	Sylvius et al. [14]
c.1526dupC	p.Thr510Tyr fsTer42	truncation	9 (2)	pathogenic	1/4	DCM, AVB, AF, nsVT, HTX	Saj et al. [15], Pugh et al. [26]
c.1549C>T	p.Gln517Ter	truncation	9 (2)	pathogenic	2/3	DCM, SCA, AVB, AF, nsVT	Stallmeyer et al. [23]
c.1621C>G	p.Arg541Gly	missense	10 (2)	likely pathogenic	1/2	DCM, nsVT	Malek et al. [22], Saj et al. [15]
c.1621C>T	p.Arg541Cys	missense	10 (2)	likely pathogenic	2/1	DCM, AVB, SCA, HTX	Forissier et al. [24], Hookana et al. [25], Saj et al. [15], Pugh et al. [26]

Legend: ACMG: American College of Medical Genetics and Genomics; AF: atrial fibrillation; AVB: atrioventricular block; DCM: dilated cardiomyopathy; HF: heart failure; HNDC: hypokinetic non-dilated cardiomyopathy; HTX: heart transplantation; ICD: implantable cardioverter-defibrillator; LVE: left ventricular enlargement; n/a: not applicable; NLS: nuclear localization signal; nsVT: non-sustained ventricular tachycardia; SCA: sudden cardiac arrest; sVT: sustained ventricular tachycardia.

**Table 2 jcm-09-01443-t002:** Baseline clinical characteristics of *LMNA* variant carriers at initial visit.

	Total *n* = 53	Probands *n* = 21 (39.6%)	Relatives *n* = 32 (60.4%)	*p*
Age (years)	33.2 ± 12.4	39.6 ± 10.0	29.0 ± 12.2	**0.002**
Men, *n* (%)	31 (58.5%)	14 (66.7%)	17 (53.1%)	0.328
*LMNA* missense variants, *n* (%)	13 (24.5%)	8 (38.1%)	5 (15.6%)	0.063
Symptoms				
Syncope, *n* (%) (*n* = 49)	12 (24.5%)	8 (42.1%)	4 (13.3%)	**0.039**
Family history of SCD <60 years, *n* (%) (*n* = 50)	25 (50%)	8 (44.4%)	17 (53.1%)	0.556
Heart failure, *n* (%)	19 (35.9%)	15 (71.4%)	4 (12.5%)	**<0.0001**
NYHA class ≥ 3, *n* (%)	7 (13.2%)	6 (28.6%)	1 (3.1%)	**0.012**
Arrhythmias				
Atrial arrhythmias, *n* (%) (*n* = 52)	19 (36.5%)	13 (61.9%)	6 (19.4%)	0.002
nsVT, *n* (%) (*n* = 50)	30 (60%)	19 (100%)	11 (35.5%)	**<0.0001**
SCA/sVT, *n* (%) (*n* = 50)	9 (18.0%)	7 (36.8%)	2 (6.4%)	**0.018**
CCD				
LBBB, *n* (%) (*n* = 47)	8 (17.0%)	8 (50.0%)	0 (0%)	**<0.0001**
AV block (≥1), *n* (%) (*n* = 52)	31 (59.6%)	17 (85.0%)	14 (43.7%)	**0.003**
Cardiomyopathies				
LVEF < 50%, *n* (%)	19 (35.8%)	15 (71.4%)	4 (12.5%)	**<0.0001**
LVEF (%)	50.5 ± 16.2	36.9 ± 15.7	59.4 ± 8.9	**<0.0001**
LVE > 112%, *n* (%)	28 (52.8%)	18 (85.7%)	10 (31.3%)	**0.0001**
LVEDD (mm)	53.7 ± 8.7	59.1 ± 8.2	50.2 ± 7.2	**0.0001**
Biomarkers				
CK (IU/l) (*n* = 46)	162.5 (93–291)	121 (83–253)	178 (93–458)	0.157
elevated CK, *n* (%) (*n* = 46)	13 (28.3%)	2 (10.5%)	11 (40.7%)	**0.025**
hs Troponin T (ng/L) (*n* = 42)	13.6 (7.0–23.9)	19.2 (13.4–28.9)	11.9 (5.7–19.9)	**0.018**
elevated hs Troponin T, *n* (%) (*n* = 42)	20 (47.6%)	10 (66.7%)	10 (37.0%)	0.065
NT-proBNP (pg/mL) (*n* = 42)	161.0 (72.7–683.7)	683.7 (224–1211)	84.9 (52.4–183.0)	**<0.001**
elevated NT-proBNP, *n* (%) (*n* = 42)	23 (54.8%)	14 (82.3%)	9 (36.0%)	**0.003**
Comorbidities				
Coronary artery disease, *n* (%)	2 (3.8%)	1 (4.8%)	1 (3.1%)	1.000
Hypertension, *n* (%)	6 (11.3)	1 (4.8%)	5 (15.6%)	0.384
Implantable devices				
ICD in primary PPX, *n* (%)	8 (15.1%)	5 (23.8%)	3 (9.4%)	0.204
ICD in secondary PPX, *n* (%)	8 (15.1%)	6 (28.6%)	2 (6.3%)	**0.047**
ICD/CRT-D implantation, *n* (%)	16 (30.2%)	11 (52.4%)	5 (15.6%)	**0.004**
Medication				
β-Blocker, *n* (%)	25 (47.2%)	16 (76.2%)	9 (28.1%)	**<0.001**
ACE-I or ARB, *n* (%)	21 (39.6%)	17 (81.0%)	4 (12.5%)	**<0.0001**
MRA, *n* (%)	6 (11.3%)	6 (28.6%)	0 (0%)	0.002

Legend: Number of subjects is expressed as *n* (%). Continuous variables are shown as mean ± SD or median and quartiles (Q1:25th–Q2:75th percentiles). ACE-I: angiotensin converting enzyme inhibitor; ARB: angiotensin receptor blocker; AV block: atrioventricular block; CCD: cardiac conduction defect; CK: creatine phoshokinase; CRT-D: cardiac resynchronization therapy defibrillator; hs: high sensitive; ICD: implantable cardioverter defibrillator; LBBB: left bundle branch block; LVE: left ventricular enlargement; LVEDD: left ventricular end-diastolic dimension; LVEF: left ventricular ejection fraction; MRA: mineralocorticoid receptor antagonist; nsVT: non-sustained ventricular tachycardia; NT-proBNP: N-terminal pro-brain natriuretic peptide; NYHA class: New York Heart Association functional class; PPX: prophylaxis; SCA: sudden cardiac arrest; SCD: sudden cardiac death; sVT: sustained ventricular tachycardia.

**Table 3 jcm-09-01443-t003:** Change in circulating biomarkers’ levels between baseline and last measurement.

Biomarker	Baseline	Last Measurement	Relative Change	*p*
hs Troponin T (ng/L) *n* = 32	15.1 (7.6–24.7)	17.4 (9.6–30.6)	+15.2%	0.002
NT-proBNP (pg/mL) *n* = 27	223.8 (72.7–683.7)	478.3 (86.3–1353)	+114%	0.0003

The results are shown as median and quartiles (Q1:25th–Q2:75th percentiles). Legend: hs Troponin T: highly sensitive troponin T; NT-proBNP: N-terminal prohormone brain natriuretic peptide.

**Table 4 jcm-09-01443-t004:** Clinical Characteristics of *LMNA* variant carriers at last follow-up.

	Total *n* = 53	Non-Missense *n* = 40	Missense *n* = 13	*p*
Age (years)	38.6 ± 12.5	39.6 ± 13.3	35.5 ± 9.6	0.318
Men, *n* (%)	31 (58.5%)	24 (60.0%)	7 (53.8%)	0.696
Probands, *n* (%)	21 (39.6%)	13 (32.5%)	8 (61.5%)	0.063
Symptoms				
Heart failure, *n* (%)	27 (50.9%)	18 (45.0%)	9 (69.2%)	0.129
NYHA class ≥ 3, *n* (%)	17 (32.1%)	11 (27.5%)	6 (46.2%)	0.306
Arrhythmias				
Atrial arrhythmias, *n* (%) (*n* = 52)	24 (46.2%)	20 (51.3%)	4 (30.8%)	0.199
nsVT, *n* (%) (*n* = 51)	35 (68.6%)	27 (69.2%)	8 (66.7%)	1.000
CCD				
LBBB, *n* (%) (*n* = 39)	10 (25.6%)	4 (14.3%)	6 (54.5%)	**0.017**
AV block (≥1), *n* (%)	36 (67.9%)	29 (72.5%)	7 (53.8%)	0.306
Cardiomyopathies				
LVEF < 50%, *n* (%)	22 (41.5%)	13 (32.5%)	9 (69.2%)	**0.019**
LVEF (%)	46.9 ± 17.4	51.9 ± 15.3	38.0 ± 17.9	**0.012**
LVE > 112%, *n* (%)	29 (54.7%)	19 (47.5%)	10 (76.9%)	0.064
LVEDD (mm)	54.8 ± 9.1	51.0 ± 10.5	60.1 ± 12.2	**0.016**
Biomarkers				
hs Troponin T (ng/L) (*n* = 42)	16.1 (9.9–29.8)	16.2 (9.9–31.5)	13.9 (10.4–17.1)	0.520
elevated hs Troponin T, *n* (%) (*n* = 42)	24 (57.1%)	20 (60.6%)	4 (44.4%)	0.462
NT-proBNP (pg/mL) (*n* = 41)	397.2 (85–1037)	359.3 (84–1012)	722.0 (86–2245)	0.324
elevated NT-proBNP, *n* (%) (*n* = 41)	26 (63.4%)	18 (60%)	8 (72.7%)	0.716
Implantable devices				
ICD in primary PPX, *n* (%)	26 (49.1%)	20 (50.0%)	6 (46.2%)	0.810
ICD in secondary PPX, *n* (%)	8 (15.1%)	5 (12.5%)	3 (23.1%)	0.389
ICD/CRT-D implantation, *n* (%)	34 (64.2%)	25 (62.5%)	9 (69.2%)	0.749
Events during follow-up				
Malignant ventricular arrhythmia, *n* (%)	13 (24.5%)	9 (22.5%)	4 (30.8%)	0.712
Appropriate ICD shock, *n* (%)	11 (31.4%)	7 (26.9%)	4 (44.4%)	0.416
RF ablation for VA, *n* (%)	7 (13.2%)	4 (10%)	3 (23.1%)	0.343
Cardiopulmonary resuscitation, *n* (%)	1 (1.9%)	1 (2.5%)	0 (0%)	1.000
Sudden cardiac death, *n* (%)	1 (1.9%)	1 (2.5%)	0 (0%)	1.000
End-stage heart failure, *n* (%)	14 (26.4%)	9 (22.5%)	5 (38.5%)	0.292
Heart transplantation, *n* (%)	11 (20.8%)	8 (20.0%)	3 (23.1%)	1.000
HF death, *n* (%)	3 (5.7%)	1 (2.5%)	2 (15.4%)	0.145

Legend: Number of subjects is expressed as *n* (%). Continuous variables are shown as mean ± SD or median and quartiles (Q1:25th–Q2:75th percentiles). RF ablation: radiofrequency ablation; VA: ventricular arrhythmia.

**Table 5 jcm-09-01443-t005:** Potential risk factors affecting MVA-free and HTX-free survival.

	Cumulate Incidence	*p*-Value Log-Rank	Univariable		Mutivariable	
			HR (95% CI)	*p*-Value Wald	HR (95% CI)	*p*-Value Wald
**MVA, from date of first visit**	**at 8 years of follow-up (*n* = 44)**					
Sex: Male vs. Female	24 vs. 20	0.929	1.07 (0.24–4.79)	0.929	-	
Mutation type: Missense vs. Other	25 vs. 22	0.727	0.69 (0.08–5.73)	0.729	-	
AV block: yes vs. no	42 vs. 0	**0.007**	NA		-	
nsVT: yes vs. no	37 vs. 9	**0.031**	7.38 (0.88–61.7)	0.064	-	
LVEF: <45% vs. >45%	48 vs. 16	0.100	3.32 (073–15.04)	0.120		
LVEF: <55% vs. >55%	42 vs. 11	**0.026**	**5.54 (1.02–27.78)**	**0.047**	-	
NT-proBNP: >150 vs. <150 pg/mL	67 vs. 6	**0.002**	**13.40 (1.6–112.7)**	**0.017**	**10.4** **(1.21–89.79)**	**0.010**
hsTn T: >20 vs. <20 ng/L	66 vs. 6	**0.003**	**13.16 (1.49–115.8)**	**0.020**	-	
**MVA, from date of birth**	**at 60 years (*n* = 53)**					
Status: Proband vs. Relative	65 vs. 62	0.746	1.18 (0.43; 3.29)	0.746	-	
Sex: Male vs. Female	100 vs. 41	0.087	2.61 (0.84; 8.05)	0.096	-	
Mutation type: Missense vs. Other	58 vs. 64	0.095	2.52 (0.82–7.72]	0.106	-	
**End-stage HF, from date of birth**	**at 60 years (*n* = 53)**					
Status: Proband vs. Relative	83 vs. 41	**0.034**	**3.71 (1.02; 13.47)**	**0.046**	-	
Sex: Male vs. Female	84 vs. 50	**0.004**	**5.55 (1.52; 20.27)**	**0.009**	**6.18** (1.66–23.0)	**0.007**
Mutation type: Missense vs. other	100 vs. 63	**0.041**	3.18 (0.99; 10.25)	0.052	**3.83** **(1.14–12.85)**	**0.029**

Legend: AV block: atrioventricular block; CI: confidence interval; HF: heart failure; hsTnT: high-sensitive Troponin T; HTX: heart transplantation; LVEF: left ventricular ejection fraction; MVA: malignant ventricular arrhythmia; NA: not applicable; nsVT: non-sustained ventricular tachycardia; NT-proBNP: N-terminal prohormone brain natriuretic peptide.

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
