# Peer review of "Can Circulating Cardiac Biomarkers Be Helpful in the Assessment of LMNA Mutation Carriers?"

_jcm, 2020, doi:10.3390/jcm9051443_

Round 1

Reviewer 1 Report

Fig.3 needs revisiting. You can barely tell anything from the figure. The authors should find another way to display the data. Or take it out all together and put the data in the supplementary info and make a comment in the paper.

The author should reword this to include clinical language or actions. 

  1. probands but also in more than one third of relatives, indicating that it might be a “red flag”  

  1. This discrepancy may be related to adopted definitions of early stage disease, but may also underscore the need of diagnostic vigilance (e.g. repeated 24-hour ECG examinations).24-hour ECG may defeat the purpose of looking for markers. Needs expanding. 

373-378 : I feel the statement is too strong. There are many point of care devices looking at markers beyond just end point.

Alawieh, 2019.. Tsui.. 2019.. Sörensen 2019. 

Worth mentioning POC. 

Author Response

Fig.3 needs revisiting. You can barely tell anything from the figure. The authors should find another way to display the data. Or take it out all together and put the data in the supplementary info and make a comment in the paper.

Data presented on Figure 3 reflect variability of troponin concentrations in particular LMNA mutation carriers within time. Following the suggestion of the Reviewer we moved the Figure to the Supplementary files and left the comment and Table 3 in the manuscript (lines240-248).

The author should reword this to include clinical language or actions. 

  1. probands but also in more than one third of relatives, indicating that it might be a “red flag”

We added the following sentence to the text, lines 321-323:

 In 2013 Rapezzi et al.[31] proposed a framework for the clinical approach to diagnosis in cardiomyopathies based on the recognition of diagnostic ”red flags” that can be used to guide rational selection of specialized tests.

  1. This discrepancy may be related to adopted definitions of early stage disease, but may also underscore the need of diagnostic vigilance (e.g. repeated 24-hour ECG examinations).24-hour ECG may defeat the purpose of looking for markers. Needs expanding. 

 Of note, the Cardiomyopathy Registry of the EURObservational Research Programme of the European Society of Cardiology showed that Holter 24hour ECG monitoring was performed only in 469/1260 (37.2%) patients with DCM and in the setting of cardiolaminopathy it is of crucial importance to monitor asymptomatic LMNA mutation carriers with Holter 24hour ecg monitoring, however the presence of abnormal TnT concentrations as shown in our study precedes the appearance of arrhythmia.

We added an analogous sentence in the main text, lines 340-345.

373-378 : I feel the statement is too strong.

We modified the statement slightly (line 384: The utility of cardiac troponins in determining prognosis in HF and DCM was established shown…

There are many point of care devices looking at markers beyond just end point.

Alawieh, 2019.. Tsui.. 2019.. Sörensen 2019. 

Worth mentioning POC.

We added a sentence in the main text, lines 389-391:

Also, due to recent advances in cardiac biomarker sensing technologies, point-of-care (POC) applications can be hopefully applied in the management of cardiomyopathies in the foreseeable future.

Reviewer 2 Report

Authors present a biomarker study on patients with laminopathy. Methods and language are sound. The manuscript contains detailed description of the studied cases and the topic is of interest.

Minor remarks:

  • Please re-check reference list. It ends with #1.
  • Table 2: the term "dysrhythmia" is not common, change to "arrhythmia"
  • Table 2: Medication: guideline-directed heart failure therapy includes MR antagonists for patients with reduced LVEF. Please provide the numbers. Were there any patients on ARNI?
  • All figures (especially fig 2,3,5,6,7) are very hard to read due to small font size or too many panels. This could be revised to increase readability. Especially the Kaplan-Meyer-graphs (fig 6+7)are nearly indecipherable.

Author Response

Authors present a biomarker study on patients with laminopathy. Methods and language are sound. The manuscript contains detailed description of the studied cases and the topic is of interest.

 Minor remarks:

  • Please re-check reference list. It ends with #1.

Corrected.

  • Table 2: the term "dysrhythmia" is not common, change to "arrhythmia"

The change has been made (lines 207, Table 2, Table 4)

  • Table 2: Medication: guideline-directed heart failure therapy includes MR antagonists for patients with reduced LVEF. Please provide the numbers. Were there any patients on ARNI?

The data regarding MRA therapy have been entered into Table 2. There were no patients on ARNI at the time of initial visit or within 6 months after.

  • All figures (especially fig 2,3,5,6,7) are very hard to read due to small font size or too many panels. This could be revised to increase readability. Especially the Kaplan-Meyer-graphs (fig 6+7) are nearly indecipherable.

Figures 3 has been moved to supplementary data online, as suggested by Reviewer #1 and Figure 5 has been removed, as suggested by Reviewer #3. Figures 2, 6 and 7 have been modified for greater legibility. We reduced the number of presented K-M graphs in the basic manuscript from 6 to 4 and moved two of them to the supplementary files.

Reviewer 3 Report

This study by Chmielewski et al. investigated the chronotropic clinical characteristics and the biomarkers especially NT-pro BNP might predict the occurrence of malignant ventricular arrhythmia (MVA) events in the patients with cardiac laminopathies. This is of interesting to evaluate prognosis and therapeutic strategies for lamin A/C mutation carriers, however, this article has many issues to be solved with.

Major comments:

#1. Author described that NT-pro BNP and hs-TnT were risk factors for MVA in the patients with LMNA mutations. However, as shown in Figure 3, hs-TnT level was variable due to several conditions. So, were those biomarkers selected at the time on “initial diagnosis”? If so, please write on more detail. Moreover, cut-off values of these biomarkers are still controversial. In figure 4, the authors used >14 ng/l in ns-TnT and >125pg/ml in NT-proBNP, but in Table5, those are 20 and 150, respectively. How are these decided? Please add ROC curve to show the highest sensitivity and 1-specificity, indicating the cut-off value of these markers.

Another question is about relationship between NT-pro BNP and hs-TnT. Which is more significant?

Why don’t the author evaluate the risk factors as hs TnT by multivariate Cox Hazard analysis in End-stage EF, Table5?

#2. According to previous study, the clinical phenotype such as MVA and HF are worse and earlier in non-missense mutation than missense mutations. This study, however, showed different results. Please explain the reason.

#3. Author evaluated the survival rate by Kaplan-Meier method in Figure6,7 and Table5. However, it is not adequate to show the Kaplan-Meier analysis from date of birth. If the author use K-M analysis, comparison between high or low BNP or TnT levels at initial diagnosis and follow up periods should be considered.

#4. Table 2 shows the baseline clinical characteristics comparison between probands and relatives,  but are these only when at first clinical visit?. Please re-write and compare between the initial visit and last followed-up and compare between probands and relatives, as well as genetic difference.

#5. Author described about hypokinetic non-dilated cardiomyopathy (HNDC) and indeterminate CM in Figure2. Please explain how and why the authors use the classification. And, please evaluate the difference of outcome among these CM.

#6. Please write the numbers who could be followed up in Figure3. Were there any data about transition of NT-pro BNP?

#7. Please write the definition of relatives in Study method.

#8. Author describe this study were retrospective and prospective study. But reviewer think this as only retrospective observational study.

#9. Figure 5 shows the time course of ICD, but not necessary for this study object.

#10. In table 4, patients with non-missense variants show lower LVEF and larger LVDD compared to those with missense variants, which may be because non-missense type is much severer LV remodeling than missense type. However, the result is completely opposite. Please explain the reason why.

Minor comments:

#1. Author described about the follow-up duration as mean in abstract part. This cohort of duration was, however, not normal distribution. Please rewrite mean to median duration.

Reviewer 4 Report

Authors performed a study focused on biomarkers for DCM patients.

Study is well designed, performed and written.

Only minor points:

1.- last reference (number should be 40 and not 1).

2.- please include data if any other rare variant was identified in any of patients analyzed using NGS technology.

3.- differences between missense and other type of mutations is clearly explained. Any difference between group of no-missense mutations (non-sense vs. frameshift)?

Round 2

Reviewer 3 Report

This revised manuscript has been well answered to the reviewer's last questions and comments, and demonstrated important clinical findings, hsTnT and NT-proBNP levels are able to predict prognosis of LMNA patients.

Although this is a single center cohort with a limited number of patients, this paper may help us to treat those refractory cardiomyopathy patients.